# A Sequential Approach to Mild Distributions

**Hans G. Feichtinger** 

Faculty of Mathematics, University of Vienna, Vienna 1090, Austria; hans.feichtinger@univie.ac.at

**Abstract:** The Banach Gelfand Triple $(S_0, L^2, S_0')(\mathbb{R}^d)$ consists of $(S_0(\mathbb{R}^d), \|\cdot\|_{S_0})$, a very specific *Segal algebra* as algebra of test functions, the Hilbert space $(L^2(\mathbb{R}^d), \|\cdot\|_2)$ and the dual space $S_0'(\mathbb{R}^d)$, whose elements are also called *"mild distributions"*. Together they provide a universal tool for Fourier Analysis in its many manifestations. It is indispensable for a proper formulation of *Gabor Analysis*, but also useful for a distributional description of the classical (generalized) Fourier transform (with Plancherel's Theorem and the Fourier Inversion Theorem as core statements) or the foundations of Abstract Harmonic Analysis, as it is not difficult to formulate this theory in the context of locally compact Abelian (LCA) groups. A new approach presented recently allows to introduce $(S_0(\mathbb{R}^d), \|\cdot\|_{S_0})$ and hence $(S_0'(\mathbb{R}^d), \|\cdot\|_{S_0'})$, the space of "mild distributions", without the use of the Lebesgue integral or the theory of tempered distributions. The present notes will describe an alternative, even more elementary approach to the same objects, based on the idea of completion (in an appropriate sense). By drawing the analogy to the real number system, viewed as infinite decimals, we hope that this approach is also more interesting for engineers. Of course it is very much inspired by the Lighthill approach to the theory of tempered distributions. The main topic of this article is thus an outline of the sequential approach in this concrete setting and the clarification of the fact that it is just another way of describing the Banach Gelfand Triple. The objects of the extended domain for the Short-Time Fourier Transform are (equivalence classes) of so-called *mild Cauchy sequences* (in short **ECmiCS**). Representatives are sequences of bounded, continuous functions, which correspond in a natural way to mild distributions as introduced in earlier papers via duality theory. Our key result shows how standard functional analytic arguments combined with concrete properties of the Segal algebra $(S_0(\mathbb{R}^d), \|\cdot\|_{S_0})$ can be used to establish this natural identification.

**Keywords:** tempered distribution; sequential approach; mild distribution; Banach Gelfand Triple; Feichtinger's algebra; Fourier transform; w*-convergence; Short-Time Fourier Transform

## 1. Introduction

It is the purpose of this article to present the vector space of "mild distributions" as the natural object of all "signals which have a bounded spectrogram" (or STFT: Short-Time Fourier Transform). We will do this in an elementary way, inspired by the classical sequential approach to distributions, in the spirit of Lighthill ([1]) or Bracewell ([2], Chap.5), which is also popular among engineers.

The basic principle will be the idea of *completion*, which allows to enlarge an object by adding elements which make it "more complete". This is a well-known concept familiar from the introduction of the real number system: The field $\mathbb{Q}$ of rational numbers is great when it comes to multiplication and even addition can be realized in a purely algebraic way, but unfortunately it shows some incompleteness, since obviously there is no rational number $q$ such that $q^2 = 2$.

In order to enlarge $\mathbb{Q}$ and create a complete field, to be named the real number system $\mathbb{R}$, one has several options, including Dedekind's section, or the very concrete idea of introducing the *infinite decimal expressions*, endowed with appropriate computational rules, which turns $\mathbb{R}$ into a field which

is complete with respect to the Euclidean distance, i.e., where every Cauchy sequence is convergent. Just recall:

**Definition 1.** *A sequence $(q_n)_{n \geq 1}$ in $\mathbb{Q}$ is called a Cauchy-sequence whenever relative distances of elements of the sequence tend to zero whenever the labels of two elements are both large enough. In symbols:*

$$\forall \varepsilon > 0 \; \exists n_0 \in \mathbb{N} \text{ such that } m, n \geq n_0 \Rightarrow |q_n - q_m| < \varepsilon.$$

Note that it is of course enough to allow rational numbers $\varepsilon \in \mathbb{Q}$ or only the choice $\varepsilon = 1/n$ for some $n \in \mathbb{N}$.

It is a standard fact that the field $\mathbb{Q}$ is not complete (with respect to the metric $d(q, q') = |q - q'|$, because one can find a Cauchy sequence with $\lim_{n \to \infty} q_n^2 = 2$, but without finding a limit $q_0 \in \mathbb{Q}$ (which then would be a rational number with $q_0^2 = 2$, which easily leads to a contradiction).

So instead of looking at $\mathbb{Q}$ itself one adds (as a replacement to the non-existent limits) the collection of Cauchy-sequences representing the same (not yet existent) limit, in order to get a complete metric space, i.e., a larger object that contains $\mathbb{Q}$ as a dense subspace, and such that within that larger object every Cauchy sequence has a limit. Since many different Cauchy-sequences "represent the same real number" one has to choose either a fixed representation (known as the representation of real numbers as infinite decimals), or one introduces equivalence classes of rational numbers, based on the following equivalence relation:

**Definition 2.** *Two Cauchy-sequences $(q_n)_{n \geq 1}$ and $(p_k)_{k \geq 1}$ are called equivalent if one can mix them in an arbitrary fashion and still have a Cauchy sequence, or in other words, if for $\varepsilon > 0$ there exists $n_1 \in \mathbb{N}$ such that*

$$n, k \geq n_1 \Rightarrow |q_n - p_k| < \varepsilon.$$

One then goes on and shows that this is indeed an equivalence relation on the vector space of Cauchy sequences (with natural element-wise operations), and that one can transfer all the structure (like addition, multiplication, inverse elements, absolute part) to this larger object, just called $\mathbb{R}$.

It is also important to note that every element $q \in \mathbb{Q}$ defines an equivalence class, via the constant sequence: $q_n = q, \forall n \geq 1$. One has positive elements in $\mathbb{R}$ and the absolute value is again well defined. Moreover, using the standard diagonalization trick one can show that every Cauchy-sequence in this new object (it is a rather tricky object seen in this way) has in fact a limit within $\mathbb{R}$, and hence every positive $r \in \mathbb{R}$ has roots of a (unique positive) square root, i.e., symbols like $\sqrt{2}$ make sense. In fact, this achievement of a (more or less unique) completion of $(\mathbb{Q}, |\cdot|)$ is a beautiful cornerstones of mathematical analysis.

Of course this is nothing but a wordy description (one of various possible) of the field of real numbers $\mathbb{R}$, with the same operations as in $\mathbb{Q}$, just by natural extension.

If one takes a somewhat wider perspective one realizes that the decimal system is a very special system. One might use other number systems, or continued fractions, but as a striking insight (usually explained in real analysis courses) any of the completions obtained gives the same object, resp. provides just an alternative description of $\mathbb{R}$, the natural completion of $\mathbb{Q}$ (with Euclidean metric).

The most abstract version of the underlying abstract principle allows to embed any *metric space* isometrically into a complete metric space, as a dense subspace. In other words, one can create a situation, which is perfectly analogue to the situation of the embedding $\mathbb{Q} \hookrightarrow \mathbb{R}$: Clearly rational numbers, by definition expressed as quotients of the form $p/q$, with $p, q \in \mathbb{N}, q \neq 0$, have to be identified with decimal expressions, in the usual way, as we have learnt in school. Once we have understood that $3/4$ is the same as $0.75$ we do not care whether $3/4 \cdot 6/5 = 18/20 = 9/10$ has been computed within the rational numbers or as $0.75 \cdot 1.2 = 0.75 + 0.15 = 0.9$. Moreover, $\mathbb{Q}$ is dense in $\mathbb{R}$, because obviously for any $\varepsilon > 0$ there exists some $k \in \mathbb{N}$ such that $(1/10)^k < \varepsilon$ and hence the finite decimal expression for $x \in \mathbb{R}$ which keeps the first $k$ digits of $x$ will provide an $\varepsilon$-approximation to $x$.

Generally, the method of completion consists of several steps:

1. First one considers *all possible Cauchy-sequences*;
2. Then one forms *equivalence classes* of Cauchy-sequences, which are the objects of the new space;
3. The *distance of the new elements* can be defined in a natural way, as the limit of the distances in the all possible Cauchy-Sequences (CS) generating the equivalence classes, and of course this does not depend on the representative of the class;
4. Any element of the *original space defines a constant sequence* which is a CS, for trivial reasons. The claimed natural embedding mapping assigns simply every element in the given space the corresponding constant equivalence class;
5. Then one verifies that this *natural embedding* of the original metric space into the new one *is isometric*, so that henceforth the copy with the new object arising from the original objects can be identified with the elements of the original space. (For the case of $\mathbb{Q} \hookrightarrow \mathbb{R}$ we recognize the rational numbers among all the infinite decimal expresss as those which are periodic, for a suitably chosen period.)
6. If further structure is available, i.e., if we have a normed space or a normed algebra, the resulting complete object is then a Banach space or a Banach algebra.

## 2. The Short-Time Fourier Transform STFT

First we introduce the *Short-Time Fourier Transform* (STFT), or *Sliding Window Fourier Transform* for continuous functions $f$ and compactly supported *window g*, using simply Riemann integrals. In the traditional approach to *Gabor Analysis* (see [3]) one starts by assuming that the signal $f$ to be analyzed and the window $g$ used for localization are in $L^2(\mathbb{R}^d)$, but from a practical viewpoint the non-symmetric assumption appears to be more natural.

Our signal space is $C_b(\mathbb{R}^d)$, the set of bounded, continuous, (real or ) complex-valued functions on $\mathbb{R}^d$. Since we do not request any decay or summability or decay conditions on $f$ we assume further that the *window function* $g \in C_c(\mathbb{R}^d)$, is a continuous (real-valued) function with compact support (i.e., for some $R > 0$ one has $g(x) = 0$ for $|x| \geq R$), and is thus Riemann integrable, with

$$\|g\|_1 := \int_{\mathbb{R}^d} |g(x)| dx < \infty.$$

**Definition 3.** *Given $f \in C_b(\mathbb{R}^d)$ and $g \in C_c(\mathbb{R}^d)$ one defines the STFT of the signal/function $f$ with respect to the window $g$ as the following function, whose argument are points of the $2d$-dimensional phase space, namely pairs $(t, s)$, with t from the so-called time-domain and s from the frequency domain, $t, s \in \mathbb{R}^d$.*

$$V_g(f)(t,s) := \int_{\mathbb{R}^d} f(x) g(x - t) e^{-2\pi i s \cdot x} dx. \tag{1}$$

For the case $d = 1$ the absolute value $|V_g f(t,s)|$ represents the frequency content of an audio-signal at time $t$ and at frequency $s$, comparable to a graphical score. A quite realistic idea of what it is can be obtained by watching the web-page www.gaborator.com, where you can even upload your own piece of music (in the standard WAV-format for audio files).

It is easy to show that the STFT of a bounded and continuous function is a bounded and continuous function of two variables:

**Lemma 1.** *Given $f \in C_b(\mathbb{R}^d)$ and $g \in C_c(\mathbb{R}^d)$. Then $V_g(f) \in C_b(\mathbb{R}^d \times \widehat{\mathbb{R}}^d)$ and*

$$\|V_g(f)\|_\infty \leq \|g\|_1 \|f\|_\infty. \tag{2}$$

*In other words, for any fixed $g \in C_c(\mathbb{R}^d)$ the mapping $f \mapsto V_g(f)$ is a continuous embedding from $(C_b(\mathbb{R}^d), \|\cdot\|_{C_b(\mathbb{R}^d)})$ into $(C_b(\mathbb{R}^{2d}), \|\cdot\|_{C_b(\mathbb{R}^{2d})})$.*

**Proof.** The obvious estimate of $V_g(f)(t,s)$ comes from

$$|V_g(f)(t,s)| \leq \int_{\mathbb{R}^d} |f(x-t)||g(x)|dx \leq \|f\|_\infty \|g\|_1.$$

The (uniform) continuity in the time direction comes from

$$|V_g(f)(t',s) - V_g(f)(t',s)| \leq \|f\|_\infty \|g - T_{t-t'}g\|_1 < \varepsilon$$

whenever $|t - t'| < \delta$, due to the uniform continuity of $g$. Finally (also uniform) continuity in the frequency direction comes in: Given $\varepsilon > 0$ one can find some $\delta > 0$ such that $|s - s'| < \delta$ implies

$$|e^{2\pi i s \cdot x} - e^{2\pi i s' \cdot x}| = |1 - e^{2\pi i (s-s') \cdot x}| < \varepsilon$$

whenever $x \in \text{supp}(g)$, which in turn implies

$$|V_g(f)(t,s) - V_g(f)(t,s')| \leq \|f\|_\infty \|g\|_1 \cdot \varepsilon.$$

The injectivity of $f \mapsto V_g(f)$ is an easy exercise.　□

It is our goal to show that in some sense the space $S_0'(\mathbb{R}^d)$ of mild distributions on $\mathbb{R}^d$ can be viewed as a *kind of completion* of $C_b(\mathbb{R}^d)$. Alternatively it can be viewed as the "largest natural vector space of signals" which have a bounded spectrogram.

In the rest of this paper we will show how to extend the domain of the STFT to a larger space of signals, which we call "mild distributions", making use of (mild) Cauchy sequences, but also by verifying that this approach provides the user with just an alternative approach to the Banach space $(S_0'(\mathbb{R}^d), \|\cdot\|_{S_0'})$, which has been introduced long ago and which has been used in a series of papers over many years. It also constitutes the outer layer of the so-called Banach Gelfand Triple $(S_0, L^2, S_0')$.

## 3. The Usual Approach to $\left(S_0(\mathbb{R}^d), \|\cdot\|_{S_0}\right)$

For mathematicians familiar with the theory of tempered distributions as it is taught in many courses one can describe the Banach Gelfand Triple $(S_0, L^2, S_0')(\mathbb{R}^d)$, consisting of the Segal algebra $\left(S_0(\mathbb{R}^d), \|\cdot\|_{S_0}\right)$, the Hilbert space $\left(L^2(\mathbb{R}^d), \|\cdot\|_2\right)$ and the dual space $(S_0'(\mathbb{R}^d), \|\cdot\|_{S_0'})$ in the following way, using the *extended STFT*, well defined for any $\sigma \in \mathcal{S}'(\mathbb{R}^d)$ (consistent with the classical definition) via

**Definition 4.** *Given any non-zero (real-valued) $g \in \mathcal{S}(\mathbb{R}^d)$ we define*

$$V_g(\sigma)(t,s) = \sigma(M_{-s}T_t g), \quad t,s \in \mathbb{R}^d.$$

Here we use the standard notations familiar from time-frequency analysis, namely $[T_z g](x) = g(x-z)$ and $[M_s h](x) = e^{2\pi i s \cdot x} h(x)$, where $s \cdot x = \langle s, x \rangle = \sum_{k=1}^d s_k x_k$ is the usual scalar product in $\mathbb{R}^d$.

Then the basic facts concerning the triple $(S_0, L^2, S_0')(\mathbb{R}^d)$ can be summarized as follows (see [3–5]):

**Theorem 1.** *For fixed $0 \neq g \in \mathcal{S}(\mathbb{R}^d)$ one has:*

- *For $f \in L^2(\mathbb{R}^d)$ one has $V_g(f) \in L^2(\mathbb{R}^{2d})$ and*

$$\|V_g(f)\|_{L^2(\mathbb{R}^{2d})} = \|g\|_2 \|f\|_2 \quad f, g \in L^2(\mathbb{R}^d).$$

- *A function $f \in L^2(\mathbb{R}^d)$ belongs to $S_0(\mathbb{R}^d)$ by definition, if $V_g(f) \in L^1(\mathbb{R}^{2d})$, and*

$$\|f\|_{S_0} := \int_{\mathbb{R}^d \times \widehat{\mathbb{R}}^d} |V_g(f)(t,s)| \, dt ds.$$

  *Any function $f \in S_0(\mathbb{R}^d)$ is continuous, bounded and absolutely Riemann-integrable;*
- *A tempered distribution $\sigma \in \mathcal{S}'(\mathbb{R}^d)$ belongs to $S_0'(\mathbb{R}^d)$ if and only if*

$$\|V_g(\sigma)\|_\infty = \sup_{(t,s) \in \mathbb{R}^d \times \widehat{\mathbb{R}}^d} |V_g(\sigma)(t,s)| < \infty.$$

  *Moreover, this expression defines an equivalent norm on $(S_0'(\mathbb{R}^d), \|\cdot\|_{S_0'})$.*

The (continuous) embedding of $(L^p(\mathbb{R}^d), \|\cdot\|_p)$ (for any $p \in [1,\infty]$) into $(S_0'(\mathbb{R}^d), \|\cdot\|_{S_0'})$ is realized via

$$h \mapsto \sigma_h(f) := \int_{\mathbb{R}^d} f(x)h(x)dx, \quad f \in S_0(\mathbb{R}^d). \tag{3}$$

Aside from the norm convergence in $(S_0'(\mathbb{R}^d), \|\cdot\|_{S_0'})$ (resp. uniform convergence of the corresponding STFTs $V_g(\sigma_n)$ it is important to discuss the so-called $w^*$-convergence of sequences. Due to the separability of $(S_0(\mathbb{R}^d), \|\cdot\|_{S_0})$ it is enough to use sequences, while in general one should use *nets* for a description of the $w^*$-topology!

**Definition 5.** *A sequence $(\sigma_n)_{n \geq 1}$ is $w^*$-convergent in $S_0'(\mathbb{R}^d)$ to $\sigma_0$ if and only if*

$$\lim_{n \to \infty} \sigma_n(f) = \sigma_0(f), \quad \forall f \in S_0(\mathbb{R}^d). \tag{4}$$

As illustration let us give a few examples as they appear in (engineering) books on Fourier Analysis, involving typically some "hand-waving" argument:

1. Dirac sequences, obtained by compression of an integrable function;

$$w^*\text{-}\lim_{\rho \to 0} \text{St}_\rho(g) = \left( \int_{\mathbb{R}} g(x)dx \right) \gamma_0,$$

  where $[\text{St}_\rho g](x) = \rho^{-d} g(x/\rho)$.
2. Riemannian sums converging to the integral, e.g. for $f \in S_0(\mathbb{R}^d)$:

$$\lim_{\alpha \to 0} \langle \alpha^d \sum_{k \in \mathbb{Z}^d} \delta_{\alpha k}, f \rangle = \lim_{\alpha \to 0} \alpha^d \sum_{k \in \mathbb{Z}^d} f(\alpha k) = \int_{\mathbb{R}^d} f(x)dx = \langle \mathbf{1}, f \rangle.$$

3. For any $f \in L^1(\mathbb{R}^d)$ the periodic versions of that function converge to the original function (this is often used to motivate the form of the Fourier integral for non-periodic functions):

$$w^*\text{-}\lim_{p \to \infty} \sum_{k \in \mathbb{Z}^d} T_{pk} f = f.$$

## 4. Mild Cauchy Sequences

We will take the space $C_b(\mathbb{R}^d)$ as a starting point, a vector space of "decent signals", where the usual vector space operations (addition, linear combinations etc.) are well defined. It is a normed vector space with respect to ordinary addition and scalar multiplication of continuous functions. The expression

$$\|f\|_\infty := \sup_{r \in \mathbb{R}^d} |f(r)| \tag{5}$$

defines the natural norm on this space, turning $(C_b(\mathbb{R}^d), \|\cdot\|_\infty)$ into a Banach space, in fact, it is even a Banach algebra with respect to pointwise multiplication, since

$$\|f \cdot g\|_\infty \leq \|f\|_\infty \cdot \|g\|_\infty, \quad f, g \in C_b(\mathbb{R}^d). \tag{6}$$

Occasionally we will need $(C_0(\mathbb{R}^d), \|\cdot\|_\infty)$, the closed ideal of function in $C_b(\mathbb{R}^d)$ which vanish at infinity, i.e., satisfy

$$\lim_{|x| \to \infty} |f(x)| = 0. \tag{7}$$

The Riesz Representation Theorem justifies to simply identify the dual space of $(C_0(\mathbb{R}^d), \|\cdot\|_\infty)$ with the space $(M_b(\mathbb{R}^d), \|\cdot\|_{M_b})$ (of bounded, regular Borel measures). The continuity of such a functional in the form

$$\mu : f \mapsto \int_{\mathbb{R}^d} f(x)d\mu(x),$$

corresponds to the usual $\sigma$-additivity used in measure theory, and the functional norm is the same as the total variation (see [6]).

Since we are interested in extending the concept of an STFT in an *elementary way* from $C_b(\mathbb{R}^d)$ to some larger space (yet to be defined). We first look at a variant of the concept of a Cauchy sequence which appears to be appropriate in the current context.

**Definition 6.** *Fix any non-zero $g \in C_c(\mathbb{R}^d)$ with $\hat{g} \in L^1(\mathbb{R}^d)$. A sequence $(h_n)_{n \geq 1}$ in $C_b(\mathbb{R}^d)$ is called a mild Cauchy sequence if the sequence $\|V_g(h_n)\|_\infty$ is bounded and*

$$\left(V_g(h_n)(s,t)\right)_{n \geq 1} \tag{8}$$

*is a Cauchy-sequence (with respect to n) for every pair $(t,s) \in \mathbb{R}^d \times \widehat{\mathbb{R}}^d$.*

Since the field $\mathbb{C}$ of complex numbers is itself complete this is of course equivalent to the assumption that the limit of such a sequence exists, i.e., for each $(t,s) \in \mathbb{R}^d \times \widehat{\mathbb{R}}^d$ one has a pointwise limit

$$\exists H(s,t) = \lim_{n \to \infty} V_g(h_n)(s,t) \tag{9}$$

and of course $H$ is then a bounded function with

$$\|H\|_\infty \leq \sup_{n \geq 1} \|V_g(h_n)\|_\infty. \tag{10}$$

**Remark 1.** *One can show that the condition does not depend on the a particular choice of g. In fact, even a slightly stronger claim is true: if condition (8) is valid for one nonzero function $g_1 \in S_0(\mathbb{R}^d)$ it is also true for any other function $g_2 \in S_0(\mathbb{R}^d)$. This follows from the atomic characterization of $(S_0(\mathbb{R}^d), \|\cdot\|_{S_0})$, a kind of exchange principle, (see [4,5]).*

We define *equivalence of Cauchy-sequences* in a rather obvious way:

**Definition 7.** *Two mild Cauchy-sequences $(h_n^{(1)})$ and $(h_k^{(2)})$ are called mildly equivalent if they have the same limit, i.e.,*

$$\lim_{n \to \infty} V_g(h_n^{(1)})(s,t) = \lim_{k \to \infty} V_g(h_k^{(2)})(s,t), \quad \forall (t,s) \in \mathbb{R}^d \times \widehat{\mathbb{R}}^d. \tag{11}$$

We also define a norm on the vector space of equivalence classes of mild Cauchy-sequences (short: ECmiCS):

**Definition 8.** *For a given ECmiCS* **F** *we define its norm:*

$$\|\mathbf{F}\|_{CS} = \inf\{\sup_{n \geq 1} \|V_g(h_n)\|_\infty\}, \tag{12}$$

*where the infimum is taken over all representatives of the equivalence class* **F***.*

It is then not difficult (but lengthly) to verify that this is actually a norm on the vector space of equivalence classes, and that $(C_b(\mathbb{R}^d), \|\cdot\|_\infty)$ is continuously embedded into this space. We will observe later that $C_b(\mathbb{R}^d)$ is *not dense* in the new space with respect to this norm, but only in the natural topology (corresponding to the $w^*$-convergence in $S_0'(\mathbb{R}^d)$). What is less obvious (but a valid claim) is the *completeness of this new space*: every mild Cauchy sequence has a limit.

In order to establish appropriate terminology for discussion and later reference let us introduce the acronym **ECmiCS** for an **E**quivalence **C**lass if **mi**ld **C**auchy **S**equences. These ECmiCS constitute the new (enlarged) vector space of objects, in fact a normed space with respect to the *CS*-norm. For the rest of this note it will be convenient to use the symbol **F** for such an equivalence class, hence $\|\mathbf{F}\|_{CS}$ describes the norm of an ECmiCS.

Using this terminology our observations reduce to the following statement:

**Lemma 2.** *The space* $(C_b(\mathbb{R}^d), \|\cdot\|_\infty)$ *is continuously embedded into the space of ECmiCS, endowed with the CS-norm.*

One only has to check (and this is quite easy) that if $h \in C_b(\mathbb{R}^d)$ represents the zero ECmiCS that it has to be the zero-function, but this is quite clear because it implies that $\int_{\mathbb{R}^d} f(x)h(x)dx = 0$ for any $f \in S_0(\mathbb{R}^d)$, again based on the atomic characterization of $(S_0(\mathbb{R}^d), \|\cdot\|_{S_0})$.

What is perhaps more interesting is the fact that one can extend more or less all the usual manipulations to this *enlarged space*, by just applying it to the individual mild Cauchy sequences, by verifying that they are compatible with the introduced equivalence relation.

Above all we have translation, modulation, dilations, which can be easily transferred to the extended space. The Fourier transform is a bit more tricky.

The decisive formula is (3.10) in [3], p. 40, also called *fundamental formula of time-frequency analysis* there.

$$V_{\widehat{g}}(\widehat{f})(\omega, -t) = e^{2\pi i t \cdot \omega} V_g(f)(t, \omega), \quad t, \omega \in \mathbb{R}^d. \tag{13}$$

For the standard choice $g = g_0$ (Gaussian window) this implies that the STFT of $\widehat{f}$ is (up to some harmless phase factors) just the (absolute value of the) original STFT of $f$, rotated by 90 degrees in phase space, thus implying the isometric invariance of $(S_0(\mathbb{R}^d), \|\cdot\|_{S_0})$ under the Fourier transform. Since at this point we do not have a Fourier transform for $h \in C_b(\mathbb{R}^d)$ we cannot use of this formula yet.

## 5. The Functional Analytic Viewpoint

In this section we will demonstrate that the approach chosen in the section above is in fact equivalent to the one used in various publications on the subject so far, or the approach to the subject possible in the context of tempered distributions. Of course, it is necessary to make use of result from standard linear *functional analysis* in order to carry out these identifications in a mathematically correct way.

Let us first recall in some more detail the characterization of $(S_0'(\mathbb{R}^d), \|\cdot\|_{S_0'})$ (as e.g., introduced in [7]) as a subspace of tempered distributions $\mathcal{S}'(\mathbb{R}^d)$:

**Lemma 3.** *The space* $S_0'(\mathbb{R}^d)$ *coincides with the space of all tempered distributions with a bounded STFT (with respect to any non-zero Schwartz function g):*

$$V_g(\sigma)(t, s) = \sigma(M_{-s}T_t g), \quad t, s \in \mathbb{R}^d.$$

**Proof.** First of all let us remind of the fact that $\boldsymbol{\mathcal{S}}(\mathbb{R}^d)$ is dense in $\big(\boldsymbol{S}_0(\mathbb{R}^d),\|\cdot\|_{\boldsymbol{S}_0}\big)$. Hence $\boldsymbol{S}_0'(\mathbb{R}^d)$ is a subspace of $\boldsymbol{\mathcal{S}}'(\mathbb{R}^d)$ (continuous embedded).

Since the set $(M_{-s}T_t g)_{t,s\in\mathbb{R}^d}$ is uniformly bounded, in $\big(\boldsymbol{S}_0(\mathbb{R}^d),\|\cdot\|_{\boldsymbol{S}_0}\big)$ due to the fact that $\|M_{-s}T_t g\|_{\boldsymbol{S}_0} = \|f\|_{\boldsymbol{S}_0}$, it is necessary for $\sigma$ to extend to a linear functional on $\big(\boldsymbol{S}_0(\mathbb{R}^d),\|\cdot\|_{\boldsymbol{S}_0}\big)$ that

$$\sup_{(t,s)\in\mathbb{R}^d\times\widehat{\mathbb{R}}^d} |V_g(\sigma)(t,s)| \le \|\sigma\|_{\boldsymbol{S}_0'}\|f\|_{\boldsymbol{S}_0}. \tag{14}$$

Conversely assume that $\sigma\in \boldsymbol{S}_0'$ is given. Then one has, thanks to the *atomic representation* of general elements of $\big(\boldsymbol{S}_0(\mathbb{R}^d),\|\cdot\|_{\boldsymbol{S}_0}\big)$ as

$$f = \sum_{n=-\infty}^{\infty} c_n M_{-s_n} T_{t_n} g \quad\text{with}\quad \sum_{n=-\infty}^{\infty} |c_n| \le C\|f\|_{\boldsymbol{S}_0}, \tag{15}$$

for some constant $C>0$ the following estimate for any $f\in\boldsymbol{\mathcal{S}}(\mathbb{R}^d)\subset\boldsymbol{S}_0(\mathbb{R}^d)$:

$$|\sigma(f)| \le \sum_{n=-\infty}^{\infty} |c_n|\sigma(M_{-s_n}T_{t_n}g)$$

and further

$$|\sigma(f)| \le \sum_{n=-\infty}^{\infty} |c_n||V_g(\sigma)(t_n,s_n)| \le C\|f\|_{\boldsymbol{S}_0}\|V_g(\sigma)\|_\infty.$$

Thus we have established the equivalence of $\|V_g(\sigma)\|_\infty$ and the standard dual norm on $\big(\boldsymbol{S}_0'(\mathbb{R}^d),\|\cdot\|_{\boldsymbol{S}_0'}\big)$, given by $\|\sigma\|_{\boldsymbol{S}_0'} := \sup_{\|f\|_{\boldsymbol{S}_0}\le 1} |\sigma(f)|$. $\square$

Since the following result will immediately provide a number of properties of our "completion" of $C_b(\mathbb{R}^d)$ we want to give it next, in order to avoid elementary, but cumbersome arguments (in the spirit of sequential approaches to generalized functions, as promoted by Lighthill [1]).

This is one of our main results:

**Theorem 2.** *There is a natural identification of $(\boldsymbol{S}_0'(\mathbb{R}^d),\|\cdot\|_{\boldsymbol{S}_0'})$ with ECmiCS, the normed space of equivalence classes of mild Cauchy sequences in $C_b(\mathbb{R}^d)$, with equivalence of norms.*

**Proof.** Given any equivalence class of mild Cauchy-sequence $\mathbf{F}$ and $\varepsilon>0$ let us choose as *mild CS* $(f_n)_{n\ge1}$ in $C_b(\mathbb{R}^d)$ with

$$\sup_{n\ge1}\|V_g(f_n)\|_\infty \le \|\mathbf{F}\|_{CS} + \varepsilon.$$

Then the sequence $(f_n)_{n\ge1}$ can be viewed as a bounded sequence $(\sigma_n)_{n\ge1}$ in $(\boldsymbol{S}_0'(\mathbb{R}^d),\|\cdot\|_{\boldsymbol{S}_0'})$, via

$$\sigma_n(f) = \int_{\mathbb{R}^d} f(x)f_n(x)dx, \quad f\in\boldsymbol{S}_0(\mathbb{R}^d).$$

In fact,

$$\sup_{n\ge1}\|\sigma_n\|_{\boldsymbol{S}_0'} \le \|\mathbf{F}\|_{CS} + \varepsilon \le 2\|\mathbf{F}\|_{CS}.$$

By the atomic decomposition formula (15) this sequence is uniformly bounded in $\boldsymbol{S}_0'(\mathbb{R}^d)$. Therefore it is enough to show that $(\sigma_n(f))_{n\ge1}$ is a Cauchy-sequence for any $f\in\boldsymbol{\mathcal{S}}(\mathbb{R}^d)$ (the existence of the limit follows then automatically). Given $\varepsilon>0$ and $f\in\boldsymbol{S}_0(\mathbb{R}^d)$ let us choose a finite sum of the form $h = \sum_{n=1}^{K} c_n M_{s_n} T_{t_n} g$

$$\sigma_r(M_{s_n}T_{t_n}g) = V_g(\sigma_r)(t_n,s_n). \tag{16}$$

with

$$\|f-h\|_{\boldsymbol{S}_0} < \varepsilon/(10\|\mathbf{F}\|_{CS}),$$

and of course $\sum_{n \geq 1} |c_n| \leq C \|f\|_{S_0}$.

Given the finite sequence $(t_n, s_n)_{n=1}^K$ in $\mathbb{R}^d \times \widehat{\mathbb{R}}^d$ the pointwise Cauchy-condition implies that we can find an index $n_1 \in \mathbb{N}$ such that one has for $r_1, r_2 \geq n_1$

$$|V_g(f_{r_1})(t_n, s_n) - V_g(f_{r_2})(t_n, s_n)| < \varepsilon \cdot (2C\|f\|_{S_0})^{-1}, \quad \text{for} 1 \leq n \leq K. \tag{17}$$

Consequently one has

$$|\sigma_{r_1}(h) - \sigma_{r_2}(h)| \leq \sum_{n=1}^K |c_n| |V_g(f_{r_1})(t_n, s_n) - V_g(f_{r_2})(t_n, s_n)| \leq \varepsilon/2. \tag{18}$$

This implies for $r_1, r_2 \geq n_1$

$$|\sigma_{r_1}(f) - \sigma_{r_2}(f)| \leq |\sigma_{r_1}(f - h)| + \varepsilon/2 + |\sigma_{r_2}(h - f)| \leq \tag{19}$$

$$\leq 2 \sup_{n \geq 1} \|\sigma_n\|_{S_0} \|f - h\|_{S_0} + \varepsilon/2 \tag{20}$$

$$\leq 2/5\varepsilon + \varepsilon/2 < \varepsilon. \tag{21}$$

This shows that there exists some $\sigma$ with $V_g(\sigma)$ being the poitnwise limit of the mild distributions $(\sigma_n)$ and thus

$$\|\sigma\|_{S_0'} = \sup |V_g(\sigma)(t, s)| \leq \|F\|_{CS} + \varepsilon,$$

but since this claim is valid for any $\varepsilon > 0$ we have

$$\|\sigma\|_{S_0'} \leq \|\mathbf{F}\|_{CS}.$$

It remains to mention (details are left to the interested reader) that the assignment $F \mapsto \sigma$ is in fact well defined, i.e., it is in fact a mapping from the *equivalence* class **F**, not depending on the representative used. But this follows from the uniqueness of the STFT: Given $\sigma_1, \sigma_2 \in S_0'(\mathbb{R}^d)$ with $V_g(\sigma_1) = V_g(\sigma_2)$ one has of course $\sigma_1 = \sigma_2$.  $\square$

Since any dual of a normed space is a complete, normed space, i.e., a Banach space, we have as an immediate consequence of the isomorphism just stated:

**Corollary 1.** *The space of equivalence classes of mild Cauchy sequences in $C_b(\mathbb{R}^d)$ with its natural norm is a Banach space. The embedding of $(C_b(\mathbb{R}^d), \|\cdot\|_\infty)$ into this space via constant sequences, i.e., by $f \mapsto (f_n)$, with $f_n = f$ for all $n \geq 1$, is continuous, with dense range (in the $w^*$-sense).*

**Proof.** Obviously the completeness of $(S_0'(\mathbb{R}^d), \|\cdot\|_{S_0'})$ transfers to the "mild completion" of $C_b(\mathbb{R}^d)$. A direct proof would be possible, by elementary means, but it would be a bit cumbersome (and less informative). The density follows from the $w^*$-density of $S_0(\mathbb{R}^d)$ in $S_0'(\mathbb{R}^d)$, combined with the inclusion

$$S_0(\mathbb{R}^d) \hookrightarrow C_b(\mathbb{R}^d) \hookrightarrow S_0'(\mathbb{R}^d).$$

$\square$

**Remark 2.** *Of course it would be possible to provide a direct proof, by means of absolutely convergent series, but this argument would much longer, but of course more elementary in terms of tools.*

Now we have to take care of the converse: Every element $\sigma \in S_0'$ defines an equivalence class of distributions, in such a way that this assignment is the inverse to the embedding just discussed.

For this purpose we will use the regularization properties which arise from the convolution relations and pointwise multiplication of Wiener amalgam spaces, which provide the following facts (see e.g., [8,9]):

**Proposition 1.** *The following inclusions are valid for the Wiener amalgam spaces relevant for the treatment of $S_0(\mathbb{R}^d) = W(\mathcal{F}L^1, \ell^1)(\mathbb{R}^d)$ and its dual space:*

1. *$W(\mathcal{F}L^1, \ell^1)(\mathbb{R}^d) * W(\mathcal{F}L^\infty, \ell^\infty)(\mathbb{R}^d) \subset W(\mathcal{F}L^1, \ell^\infty)(\mathbb{R}^d)$;*
2. *$W(\mathcal{F}L^1, \ell^\infty)(\mathbb{R}^d) \cdot W(\mathcal{F}L^1, \ell^1)(\mathbb{R}^d) \subset W(\mathcal{F}L^1, \ell^1)(\mathbb{R}^d)$;*
3. *$W(\mathcal{F}L^1, \ell^1)(\mathbb{R}^d) \cdot W(\mathcal{F}L^\infty, \ell^\infty)(\mathbb{R}^d) \subset W(\mathcal{F}L^\infty, \ell^1)(\mathbb{R}^d)$.*

*In fact, the Wiener amalgam space $W(\mathcal{F}L^\infty, \ell^\infty)(\mathbb{R}^d)$ coincides (with equivalence of norms) with the space of pointwise multipliers of $(S_0(\mathbb{R}^d), \| \cdot \|_{S_0})$.*

Using the fact which has been the original definition (in [4], see also [3] where these spaces are introduced as modulation spaces) that

$$S_0(\mathbb{R}^d) = W(\mathcal{F}L^1, \ell^1)(\mathbb{R}^d) \quad \text{and} \quad S_0'(\mathbb{R}^d) = W(\mathcal{F}L^\infty, \ell^\infty)(\mathbb{R}^d)$$

with equivalence of norms one easily combines these relations to obtain

$$[S_0'(\mathbb{R}^d) * S_0(\mathbb{R}^d)] \cdot S_0(\mathbb{R}^d) \subset S_0(\mathbb{R}^d) \quad \text{and} \quad [S_0'(\mathbb{R}^d) \cdot S_0(\mathbb{R}^d)] * S_0(\mathbb{R}^d), \tag{22}$$

of course again combined with corresponding norm estimates, stating that there exists some $C' > 0$ (depending on the choice of the norms) such as (for example)

$$\|(\sigma * g) \cdot h\|_{S_0} \leq C' \|\sigma\|_{S_0'} \|g\|_{S_0} \|h\|_{S_0}, \quad \forall \sigma \in S_0' \text{ and } \forall g, h \in S_0. \tag{23}$$

In order to create a *mild Cauchy sequence* representing $\sigma \in S_0'$ it is thus enough to make use of bounded approximate units from in $S_0(\mathbb{R}^d)$, i.e., an Dirac sequence forming a bounded approximate unit for convolution in $(L^1(\mathbb{R}^d), \| \cdot \|_1)$ (i.e. bounded with respect to the $L^1(\mathbb{R}^d)$-norm) consisting entirely of elements from the dense subspace $S_0(\mathbb{R}^d) \subset (L^1(\mathbb{R}^d), \| \cdot \|_1)$, and another bounded approximate unit for pointwise multiplication, now bounded with respect to the Fourier algebra $(\mathcal{F}L^1(\mathbb{R}^d), \| \cdot \|_{\mathcal{F}L^1})$.

Of course one can get one from the other. In particular, it is clear, that for a bounded sequence $(g_n)_{n \geq 1}$ in $(L^1(\mathbb{R}^d), \| \cdot \|_1)$ with

$$\|g_n * g - g\|_{L^1} \to 0 \text{ for } n \to \infty, \ \forall g \in L^1(\mathbb{R}^d), \tag{24}$$

then (by the convolution theorem) $h_n = \widehat{g_n}$ is a bounded sequence in $(\mathcal{F}L^1(\mathbb{R}^d), \| \cdot \|_{\mathcal{F}L^1})$ with

$$\|h_n * h - h\|_{\mathcal{F}L^1} \to 0 \quad \text{for } n \to \infty, \ \forall h \in \mathcal{F}L^1(\mathbb{R}^d), \tag{25}$$

and in fact vice versa.

The most convenient way to produce such a Dirac sequence is of course (area preserving) compression of a function $g \in S_0(\mathbb{R}^d)$ with $\int_{\mathbb{R}^d} g(x)dx = \hat{g}(0) = 1$, using the dilation operator $\text{St}_\rho$ given by

$$[\text{St}_\rho(g)](z) = \rho^{-d} g(z/\rho), \quad \rho > 0, z \in \mathbb{R}^d,$$

satisfying (24) for $\rho_n \to 0$, with

$$\|\text{St}_\rho(g)\|_{L^1} = \|g\|_{L^1}, \quad g \in L^1(\mathbb{R}^d), \rho > 0.$$

The Fourier transform of such a sequence is characterized by the "value-preserving" dilation operator $D_\rho$, given by

$$[D_\rho(h)](z) = h(\rho z), \quad \rho > 0, z \in \mathbb{R}^d,$$

with

$$\|D_\rho(h)\|_\infty = \|h\|_\infty, \quad h \in C_b(\mathbb{R}^d), \rho > 0.$$

In other words, we have the *intertwining relation*

$$\mathcal{F}(St_\rho(g)) = D_\rho(\mathcal{F}(g)), \quad g \in L^1(\mathbb{R}^d), \rho > 0. \tag{26}$$

Whenever $\int_{\mathbb{R}^d} g(x)dx = 1$ for $g \in L^1(\mathbb{R}^d)$ or equivalently $h(0) = \widehat{g}(0) = 1$ for $h \in \mathcal{F}L^1(\mathbb{R}^d)$ the corresponding families

$$(St_{\rho_n}(g))_{n\geq 1} \quad \text{and} \quad (D_{\rho_n}(h))_{n\geq 1}$$

form such approximate units for any (fixed) sequence $\rho_n \to 0$.

For our purpose it is most convenient to choose $g = g_0$, given as $g_0(t) = e^{-\pi\|t\|^2} = \prod_{j=1}^d e^{-\pi t_j^2}$, the normalized Gauss function, which belongs to $\mathcal{S}(\mathbb{R}^d) \subset S_0(\mathbb{R}^d)$ and is Fourier invariant, i.e., we have $g = g_0 = \widehat{g}_0 = h$ and our assumptions are satisfied.

Putting all these observations together we get: For any sequence $(\rho_n)_{n\geq 1}$ with $\lim_{n\to\infty}\rho_n = 0$ the sequence

$$f_n = (St_{\rho_n}g_0 * \sigma) \cdot D_{\rho_n}g_0 \tag{27}$$

or alternatively

$$\tilde{f}_n = (D_{\rho_n}g_0 \cdot \sigma) * St_{\rho_n}g_0 \tag{28}$$

describe (equivalent) mild Cauchy sequences, both representing the given $\sigma \in S_0'$.

In fact, one has (for the case of $(f_n)$) for any $(t,s) \in \mathbb{R}^d \times \widehat{\mathbb{R}}^d$:

$$V_g(f_n)(s,t) = \langle (St_{\rho_n}g_0 * \sigma) \cdot D_{\rho_n}g_0, M_{-s}T_t g \rangle = \tag{29}$$

by the evenness of $g_0$ and hence $St_{\rho_n}g_0$, and writing for the right hand for fixed $(t,s) \in \mathbb{R}^d \times \widehat{\mathbb{R}}^d$ simply $f = M_{-s}T_t g \in S_0(\mathbb{R}^d)$:

$$= \langle (St_{\rho_n}g_0 * \sigma) \cdot D_{\rho_n}g_0, f \rangle = \langle (St_{\rho_n}g_0 * \sigma), D_{\rho_n}g_0 \cdot f \rangle = \sigma(St_{\rho_n}g * [D_{\rho_n}g_0 \cdot f]).$$

Thus it is finally left to us to verify that the argument of $\sigma$ in the last expression is in fact convergent to $f$ in the norm of $(S_0(\mathbb{R}^d), \|\cdot\|_{S_0})$.

This can be derived from the fact (for any Segal algebra, see [10,11])

$$\|St_{\rho_n}g_0 * f - f\|_{S_0} \to 0, \quad \text{for } n \to \infty, \forall f \in S_0(\mathbb{R}^d), \tag{30}$$

and (by just taking the Fourier transform of this last equation and replacing $\widehat{f} \in \mathcal{F}(S_0(\mathbb{R}^d)) = S_0(\mathbb{R}^d)$ by $f$ again):

$$\|D_{\rho_n}g_0 \cdot f - f\|_{S_0} \to 0, \quad \text{for } n \to \infty, \forall f \in S_0(\mathbb{R}^d). \tag{31}$$

These two estimates can be combined to the required claim by the following chain of inequalities, using the triangular equation and the $L^1$-boundedness of the Dirac sequence $(St_{\rho_n}g)_{n\geq 1}$. Given $f \in S_0(\mathbb{R}^d)$ one has for $n$ large enough:

$$\|St_{\rho_n}g * [D_{\rho_n}g_0 \cdot f] - f\|_{S_0} \leq \|St_{\rho_n}g * [D_{\rho_n}g_0 \cdot f - f]\|_{S_0} + \|St_{\rho_n}g * f - f\|_{S_0}$$

$$\leq \|St_{\rho_n}g\|_1 \cdot \|D_{\rho_n}g_0 \cdot f - f\|_{S_0} + \varepsilon = (\|g\|_1 + 1) \cdot \varepsilon. \tag{32}$$

This establishes that each $\sigma \in S_0'$ defines an *mild Cauchy sequence*.

It is clear from the proof that of course the mild distribution arising from such a mild CS is in fact the original mild distribution, since we have

$$\lim_{n \to \infty} V_g(f_n)(t,s) = V_g \sigma(t,s), \quad \forall (t,s) \in \mathbb{R}^2. \tag{33}$$

Altogether the above considerations show that it is possible to view $(S_0'(\mathbb{R}^d), \| \cdot \|_{S_0'})$ as a kind of "natural completion" of $(C_b(\mathbb{R}^d), \| \cdot \|_\infty)$, based on the concept of *mild Cauchy sequences*, very much in the spirit of the sequential approach to distributions, as featured by Lighthill ([1]) or as pointed out in the book of Bracewell ([2], Chap.5 and Chap.6).

*Connections to Gabor Analysis*

In practice it is not possible to compute $V_g(\sigma)$ a continuous functions of $2d$ variables for uncountably many arguments $(t,s) \in \mathbb{R}^{2d}$. Hence one may ask whether it is enough to verify boundedness resp. pointwise convergence on a sufficiently large resp. dense subset of the TF-plane (phase space).

There is a precise answer to this question, closely connected to the theory of Gabor frames. Without going into details (and leaving it to the reader to consult with [3,12,13] or other more recent sources on Gabor analysis) let us summarize the most important facts. Although one could use general lattices $\Lambda = *\mathbb{Z}^d \lhd \mathbb{R}^d \times \widehat{\mathbb{R}}^d$, for suitable non-singular $2d \times 2d$-matrices we restrict our attention to lattices of the form $\Lambda = a\mathbb{Z}^d \times b\mathbb{Z}^d$, with $a, b > 0$. We call $a$ the *time-step* (or grid constant in the image domain for $d = 2$) and $b$ the *frequency-step* (or grid constant in the wave-number domain for $d = 2$).

A *(regular) Gabor family* generated by the triple $(g, a, b)$ is a family of the form $\{ M_{nb} T_{ka} g \mid n, k \in \mathbb{Z}^d \}$, for some $g \in L^2(\mathbb{R}^d)$. In a short-hand notation we will use the more abstract symbol $g_\lambda := M_{nb} T_{ka} g$ for the Gabor atom located at $\lambda = (ka, nb)$ in phase space. A regular Gabor family is a *Gabor frame* if the *Gabor frame operator*

$$S(f) := S_{(g,a,b)} f := \sum_{\lambda \in \Lambda} \langle f, g_\lambda \rangle g_\lambda \tag{34}$$

is a bounded and invertible operator on $(L^2(\mathbb{R}^d), \| \cdot \|_2)$. Then any $f \in L^2(\mathbb{R}^d)$ can be represented as a norm-convergent double sum in $(L^2(\mathbb{R}^d), \| \cdot \|_2)$, of the form

$$f = \sum_{\lambda \in \Lambda} V_{\widetilde{g}}(\lambda) g_\lambda, \tag{35}$$

where $\widetilde{g} = S^{-1}(g) = S_{(g,a,b)}^{-1} g$ is the *dual Gabor atom*, providing the minimal norm representation of $f$ using the Gabor family $(g_\lambda)_{\lambda \in \Lambda}$, with the important estimate

$$\| V_{\widetilde{g}} f|_\Lambda \|_{\ell^2(\Lambda)} \leq C \| f \|_{L^2(\mathbb{R}^d)}, \quad \forall f \in L^2(\mathbb{R}^d), \tag{36}$$

for some constant depending only on $g$ and $(a, b)$ (in fact, suitable normalized only on $\gamma_0$, i.e., the density of the lattice $\Lambda = a\mathbb{Z}^d \times b\mathbb{Z}^d$).

It is one of the most important results of Gabor Analysis that $g \in S_0(\mathbb{R}^d)$ not only implies that $S_{g,a,b}$ is a bounded operator on $(S_0(\mathbb{R}^d), \| \cdot \|_{S_0})$, due to the estimate

$$\| (V_{\widetilde{g}}(f)(\lambda))|_\Lambda \|_{\ell^1(\Lambda)} \leq C_1 \| f \|_{S_0}, \forall f \in S_0(\mathbb{R}^d),$$

but also that $\widetilde{g}$ belongs $S_0(\mathbb{R}^d)$, see [14,15].

**Lemma 4.** *Given non-zero $g \in S_0(\mathbb{R}^d)$ there exists $\gamma_0 > 0$ such that for $a \leq \gamma_0, b \leq \gamma_0$ one has: $(g, a, b)$ generates a Gabor frame, with $\widetilde{g} \in S_0(\mathbb{R}^d)$. Hence $V_{\widetilde{g}}(\sigma)$ is well defined for $\sigma \in S_0'(\mathbb{R}^d)$, and one has:*

- $f \in S_0(\mathbb{R}^d)$ *if and only if* $V_g(f)|_\Lambda \in \ell^1(\Lambda)$;
- $f \in S_0(\mathbb{R}^d)$ *if and only if* $V_{\widetilde{g}}(f)|_\Lambda \in \ell^1(\Lambda)$;

- $f \in S_0(\mathbb{R}^d)$ *if and only if   f has a representation*

$$f = \sum_{\lambda \in \Lambda} c_\lambda g_\lambda, \text{ for some sequence}(c_\lambda)_{\lambda \in \Lambda} \in \ell^1(\Lambda). \tag{37}$$

*In each case the $\ell^1$-norm of the involved coefficients (the infimum over all possible representations in the case of (37)) defines an equivalent norm on $\left(S_0(\mathbb{R}^d), \|\cdot\|_{S_0}\right)$.*

Based on this observation it is easy to verify by simple modifications of the proofs given above (the details are left to the interested reader):

**Proposition 2.** *In the situation of Lemma 4 a sequence $(h_n)_{n \geq 1}$ in $C_b(\mathbb{R}^d)$ is a mild Cauchy sequence if and only if*

$$sup_{n \geq 1} \sup_{k,l \in \mathbb{Z}^d} |V_g(h_n)(ak, bl)| < \infty$$

*and*

$$\left(V_g(h_n)(ak, kl)\right)_{n \geq 1} \tag{38}$$

*is a Cauchy-sequence (with respect to n) for any $k, l \in \mathbb{Z}^d$.*

For specific Gabor atoms $g \in S_0(\mathbb{R}^d)$ (sufficient conditions can be found in [16] or [17]) one can show that the otherwise necessary condition $a \cdot b < 1$ is also a sufficient condition for creating a Gabor frame. For a long time this was known to be a valid criterion for the Gauss function (see [18,19]). Quite recently this criterion has been extended to the class of *totally positive functions* in [20].

## 6. Natural Extension of Operators

The idea of *generalized functions* or *distributions* is not so much to discuss linear functionals, but rather treat objects which are "more general than functions" as if they where functions. In fact, one wants to include in the mathematical discussions objects which arise naturally in analysis, such as Dirac measures, Dirac combs or pure frequencies $\chi_s(t) = e^{2\pi i \langle s,t \rangle}$ which are either not ordinary (pointwise defined) functions resp. not integrable. Still, one would like to define operations like convolution, pointwise multiplication or Fourier transforms on these objects in a way which extends these operations, as known for ordinary functions, as "naturally" as possible.

Distributions can be shifted, multiplied with decent functions, they can be dilated (or even rotated for $d > 1$) and one can take a Fourier transform. It is also possible to define their support and it behaves in the expected way (e.g., with respect to translation or dilation operators), we will not discuss this in detail here.

Clearly the extension of the operators defined on ordinary functions, say at least on $S_0(\mathbb{R}^d)$ or $C_b(\mathbb{R}^d)$ to the more general setting of $S_0'(\mathbb{R}^d)$ or in our setting to *ECmiCS* should be compatible, i.e., the extended operator should always be the (only) natural operator defined on the extension of the given operator in such a way that one has for any ordinary function:

Given an ordinary function and applying the operator first and then the embedding into ECmiCS should be the same as applying the extended operator to the ECmiCS generated by the ordinary function.

In the context of the Banach Gelfand Triple $(S_0, L^2, S_0')(\mathbb{R}^d)$ one would argue that the operators should not only map the Hilbert space to the Hilbert space, but also the test functions to test functions, and finally the dual spaces into each other, not only in a norm-continuous way, but also in a $w^*$-$w^*$-continuous form. But we will not make use of this connection.

The answer to this request is for most cases as natural as it is simple. If an ECmiCS is represented by a mild Cauchy sequence $(f_n)_{n \geq 1}$ one tries to define the extension operator $\tilde{T}$ and via $(Tf_n)_{n \geq 1}$. Of

course one has to verify in concrete cases that this is well defined, i.e., maps mild Cauchy sequences into mild Cauchy sequences and preserves equivalence classes.

We will provide a short discussion of the key steps only for the case of the Fourier transform $T = \mathcal{F}$. In this case we reduce the discussion to a sequence $(f_n)_{n \geq 1}$ with $f_n \in S_0(\mathbb{R}^d)$ for $n \geq 1$.

Let us first consider the sequence $(\widehat{f}_n)_{n \geq 1}$. In order to check that it is a mild Cauchy sequence we use a Gaussian window $g = g_0$, because it has the advantage of being Fourier invariant. Then by Plancherel's Theorem (see [3], formula (3.10)):

$$V_{g_0}(\widehat{f}_n)(t,s) = \langle \widehat{f}_n, M_s T_t g_0 \rangle = \langle f_n, T_{-s} M_t g_0 \rangle = e^{-2\pi i t \cdot s} \langle f_n, M_t T_{-s} g_0 \rangle, \tag{39}$$

or in short

$$\left( V_{g_0}(\widehat{f}_n) \right)(t,s) = e^{-2\pi i t \cdot s} V_{g_0}(f_n)(-s,t), \quad (t,s) \in \mathbb{R}^d \times \widehat{\mathbb{R}}^d. \tag{40}$$

The same identities also allow to conclude that the mapping $f \mapsto \widehat{f}$ preserves equivalence classes and that of course this extension is compatible with the usual definition of an extended Fourier transform (possible for any Fourier invariant Banach space of test functions):

$$\widehat{\sigma}(f) = \sigma(\widehat{f}), \quad f \in S_0(\mathbb{R}^d), \sigma \in S_0'. \tag{41}$$

In fact, we verify that for $(t,s) \in \mathbb{R}^d \times \widehat{\mathbb{R}}^d$ one has:

$$V_{g_0}(\widehat{f})(t,s) = \langle \widehat{f}, M_s T_t g_0 \rangle = \langle f, \mathcal{F}^{-1}(M_s T_t g_0) \rangle$$

with the limit $\lim_{n \to \infty} V_{g_0}(\widehat{f}_n)(s,t)$, for any mild CS representing $\sigma$.

## 7. Alternative Starting Points

Although we consider $C_b(\mathbb{R}^d)$ as a natural starting point when it comes to describe the "largest space of signals" for which the STFT is still a bounded functions via *equivalence classes of mild Cauchy sequences* (**ECmiCS**) one may ask whether, by the same form of "completion", other starting points could be chosen, in order to get the same space, but derive more easily additional properties (like Fourier invariance). Alternatively, one might ask, whether certain choices which are closer to applications (like the use of periodic, discrete signals as point of departure) yield other objects. Fortunately this will not be the case. The discussion of these two points makes up the current section.

The first result in this direction describes a general observation, making use of the functional analytic setting:

**Proposition 3.** *Given a Banach space* $(B, \| \cdot \|_B) \hookrightarrow (S_0'(\mathbb{R}^d), \| \cdot \|_{S_0'})$ *and $w^*$-dense. Assume that there is a bounded sequence of operators* $(A_n)_{n \geq 1}$ *on* $(S_0'(\mathbb{R}^d), \| \cdot \|_{S_0'})$ *such that for any* $n \geq 1$ $A_n$ *maps* $S_0'(\mathbb{R}^d)$ *continuously into* $(B, \| \cdot \|_B)$

$$f = w^*\text{-}\lim_{n \to \infty} A_n(f), \quad \forall f \in S_0(\mathbb{R}^d). \tag{42}$$

*Then the normed space of* **ECmiCS** *arising from* $(B, \| \cdot \|_B)$ *can be naturally identified with the space of* **ECmiCS** *arising from* $(C_b(\mathbb{R}^d), \| \cdot \|_\infty)$.

**Proof.** First of all we recall that $S_0(\mathbb{R}^d)$ is $w^*$-dense in $S_0'(\mathbb{R}^d)$ and that for any given $\sigma \in S_0'$ the sequence

$$f_n = R_n(\sigma) := D_{\rho_n} g_0 \cdot [\text{St}_{\rho_n} g_0 * \sigma]$$

is bounded in $(S_0'(\mathbb{R}^d), \| \cdot \|_{S_0'})$ and tends to $\sigma$ in the $w^*$-sense. The sequence of operators $(R_n)$ is the prototype of a sequence as alluded in the assumptions of this proposition.

For the proof of our claim let us fix the Banach space $(B, \| \cdot \|_B)$ and the sequence $(A_n)_{n \geq 1}$.

Given a mild CS-sequence in $C_b(\mathbb{R}^d)$ we want to find a representative in the same equivalence class consisting of members of $B$ (instead of the original choice $C_b(\mathbb{R}^d)$).

Given thus $(f_n)_{n\geq 1}$ in $C_b(\mathbb{R}^d)$ (viewed as elements of $S_0'(\mathbb{R}^d)$) we may consider $b_n := (A_n f_n)_{n\geq 1}$. By the assumptions we have $b_n \in B$ for each $n \geq 1$. Next we verify that this is (of course) also a mild CS. Given $g \in S_0(\mathbb{R}^d)$ we fix $(t,s) \in \mathbb{R}^d \times \widehat{\mathbb{R}}^d$ and put $h = M_s T_t g$ and watch the behavior of

$$V_g(b_n)(t,s) = V_g(A_n f_n)(t,s) = A_n f_n(h)$$

for $n \to \infty$. One has

$$|A_n f_n(h) - f_n(h)| = |f_n(A_n^* h - h)| \to 0 \quad \text{for } n \to \infty.$$

This implies at once that the new sequence $(b_n)_{n\geq 1}$ is equivalent to the original one, as well as (by consequence) it is a mild Cauchy sequence itself.

By a similar argument one verifies that two equivalent sequence $(f_n)$ and $(\tilde{f}_n)$ give rise to equivalent sequences $A_n(f_n)$ and $A_n(\tilde{f})$, thus showing that the equivalence classes are preserved by the replacement.

It remains to be shown that the new version of **ECmiCS** allowing only the representatives from $B$ describes an equivalent norm, i.e., to verify that the restriction/modification in the set of representatives does not have an effect on the corresponding infimum's norm. This is quite plausible because the norm estimate only uses $\|V_{g_0}\sigma\|_\infty$ resp. $S_0'$-norms.

Concrete estimate rely on the boundedness of the family $(A_n)$ on $(S_0'(\mathbb{R}^d), \|\cdot\|_{S_0'})$:

$$\sup_{n\geq 1} \|V_g(A_n(f_n))\|_\infty = \sup_{n\geq 1} \|A_n(f_n)\|_{S_0'} \leq C' \cdot \sup_{n\geq 1} \|f_n\|_{S_0'}.$$

which implies that the corresponding inf-norms for **ECmiCS** are equivalent. $\square$

## 8. References and History

The sequential approach to distribution theory is not new. Usually described as a way to handle "generalized functions" without making (*explicit!*) use of methods from functional analysis it offers the possibility to deal with objects which cannot be treated in the context of ordinary functions or even equivalence classes of measurable functions (as they are described by the space $(L^p(\mathbb{R}^d), \|\cdot\|_p)$, for $1 \leq p \leq \infty$).

The most well known book in that direction is certainly the small booklet of Lighthill (first published in 1962), see [1]. But there are other ones following a similar path, such as Jones ([21]) or later Antosik/Mikusinki [22]. In the book by Bracewell ([2], Chap.5 and Chap.6) also some comments are made about the possibility of a sequential approach to distribution theory, avoiding the theory of topological vector-spaces and duality theory, ready to be used by engineers.

The approach starts with a definition of *good functions* and *fairly good functions*, and the observation that the Fourier transform of a good functions is again a good function (later on, that the Fourier inversion theorem applies in a pointwise sense). In the usual literature describing the Schwartz theory of *tempered distributions* good functions are called *rapidly decreasing* or also *Schwartz functions*, fairly good functions are just the pointwise multipliers of the Schwartz space (see also [23], check).

The *sequential approach* to tempered distributions then goes on the define *regular sequences* of test functions, showing that for each regular sequence of test functions $(f_n)$ in $\mathcal{S}(\mathbb{R}^d)$ the limit

$$\lim_{n\to\infty} \int_{\mathbb{R}^d} f_n(x) F(x) dx \tag{43}$$

exists, for every $F(x) \in \mathcal{S}(\mathbb{R}^d)$. Translated into the functional analytic setting a regular sequence is a sequence of *regular distributions* arising from test functions $f_n \in \mathcal{S}(\mathbb{R}^d)$ via

$$\sigma_n(F) = \int_{\mathbb{R}^d} F(x) f_n(x) dx, \tag{44}$$

which is $w^*$-convergent to some limit.

Going on to call the sequence $(f_n)$ a distribution and use the symbol $f$ for it is similar to identify an infinite decimal expression with the sequence of finite decimal approximations by truncating the (potentially infinite) sequence at some point and leaving the rest equal to zero.

In this sense the convention

$$\int_{\mathbb{R}^d} f(x) F(x) dx := \lim_{n \to \infty} \int_{\mathbb{R}^d} f_n(x) F(x) dx$$

is meaningful, but one has to be careful not to confuse the *symbolic expression* (left hand side of the above expression) with an effective integral. But the situation is not so far from the use of the symbol $1/\pi$ for the multiplicative inverse of the irrational number $\pi$, which also is quite different from the rational number $4/3$ which obviously is the multiplicative inverse to $3/4$, because in the setting of rational numbers both symbols have an a priori meaning.

Coming back to the situation described by Lighthill it is clear that there are many sequences of test functions defining the same distribution (e.g. a Dirac Delta distribution), hence one has to define a natural equivalence relation. Altogether the starting point for the sequential approach to tempered distribution is the definition of such a distribution as an *equivalence class of regular sequences of test functions*.

This approach is taken by Jones [21] and Antosik, Mikusinksi and Sikorski in [12], and of course [1], where the readers can find more details.

In order to compare the two settings let us just recall that it is long known that $\mathcal{S}(\mathbb{R}^d) \hookrightarrow (S_0(\mathbb{R}^d), \|\cdot\|_{S_0})$ (see [24]) it is plausible that *mild Cauchy sequences* of test functions are also *regular sequences*. We just have to mentiona small extra condition, which allows us to formulate the following claim properly.

**Lemma 5.**

- *For every mild CS $(h_n)$ in $C_b(\mathbb{R}^d)$ there exists an equivalent sequence $(f_n)$ in $\mathcal{S}(\mathbb{R}^d)$. In other words, every equivalence class defining a mild distribution can be constituted with the help of mild Cauchy sequences from $\mathcal{S}(\mathbb{R}^d)$.*
- *Any mild CS in $\mathcal{S}(\mathbb{R}^d)$ is also a regular sequence, which implies that any mild distribution (viewed as ECmiCS) also defines a tempered distribution (in the sense of Lighthill).*
- *A regular sequence of test functions defines a mild distribution if and only if*

$$\sup_{n \geq 1} \|V_g(f_n)\|_\infty < \infty.$$

**Proof.** The first claim can be verified by simply applying to a given sequence $(h_n)$ in $C_b(\mathbb{R}^d)$ the usual regularization operators, already used in the earlier part of this note (i.e., smoothing with a Dirac-like Gaussian and localization by pointwise multiplication with a dilated Gaussian, very much like Fourier multipliers). By tuning the parameters it is easy to verify that one can establish equivalence.

The remaining statements follow from the fact that $\mathcal{S}(\mathbb{R}^d)$ is a *dense* subspace of $(S_0(\mathbb{R}^d), \|\cdot\|_{S_0})$. Details are left to the reader.  □

**Remark 3.** *As a final remark let us note that this article is part of a series of articles aiming at a demonstration that the Banach Gelfand Triple can be viewed as a universal tool for the treatment of problem in signal processing. The connections to classical analysis are described in [25]. An alternative approach showing how to introduce*

$\left(\boldsymbol{S}_0(\mathbb{R}^d), \|\cdot\|_{\boldsymbol{S}_0}\right)$ *based on Riemann integrals is provided in [26]. The content of [27,28] show how it could be used for the teaching of engineers and physicists.*

There are of course many references to the classical approach to distribution theory of L. Schwartz (originally [29]), such as [30], see also [31].

**Funding:** This research received no external funding. Essential parts of this manuscript have been prepared while the author held a guest position at Charles University Prague (autumn of 2018).

**Acknowledgments:** The author would like to thank Mads Jakobsen and Anupam Gumber for a careful reading of the manuscript and various corrections of the original version.

**Conflicts of Interest:** The author declares no conflict of interest.

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
