# Peer review of "A Sequential Approach to Mild Distributions"

_axioms, doi:10.3390/axioms9010025_

Round 1
Reviewer 1 Report
Report on the paper \\
"A Sequential Approach to Mild Distributions" \\
by Hans G. Feichtinger
\end{center}
The Banach Gelfand triple $ (S_0, L^2, S_0 ') (\mathbb{R}^d) $, as explained for example in [8] from the list of references, is seen as a universal tool for Fourier Analysis
in its different manifestations. The present article shows an alternative approach
to the same objects, based on the idea of completion. More precisely, the main objects under the study are the equivalence classes
of so-called mild Cauchy sequences (ECmiCS). In that framework, it is shown that ECmiCS give rise to an elementary approach to the Banach Gelfand triple.
\par
The paper is well structured, the ideas are presented in a clear and natural way, and the proofs are correct and easy to follow.
The sections are ordered to support the natural flow of the main ideas of the paper, and we suggest that (the only) subsection {\em Connections to Gabor analyisis}
should become a separate section in the revised version.
\par
The main aim of the paper is well explained in its abstract, and the introduction offers a reminder of the most
common completion procedure, i.e. the realization of the reals as equivalence classes of Cauchy sequences of rational numbers.
Section 3 offers the definition of the main tool used in the sequel, namely the sort-time Fourier transform (STFT),
and the usual approach to the Banach space $ (S_0 (\mathbb{R}^d), \| \cdot \|_{S_0} ) (\mathbb{R}^d) $ is given in Section 4. The main notions of the paper, namely the mild Cauchy sequences and their equivalence classes are observed in Section 5. Section 6 contains the exposition of the main result of the paper, i.e. the identification of the Banach Gelfand triple within the aforementioned completion procedure. Connections to Gabor analysis are briefly discussed in subsection 6.1 and the natural extension of operators is explained in Section 7. In Section 8 a discussion related to an alternative starting point is offered, while the concluding Section 9 contains the author's
notes on references and history of sequential approach to distribution theory.
\par
Below we list specific comments which should be taken into account during the (mild) revision of the paper.
\begin{enumerate}
\item page 6, Theorem 1, third item: please add "cf. Lemma 3." before {\em Moreover, this expression....}
\item page 6, after the equation (3): maybe a remark on embedding of $S_0$ into $ L^p $ can be mentioned, since the embedding of $L^p$ into $S_0 ' $ is discussed there above.
\item page 7, line 2: please explain $\gamma_0$?
\item page 9, line 3: instead of "is continuously embedded into this space." we suggest "is continuously embedded into this space, see Corollary 1."
\item page 9, paragraph after Lemma 2: instead of "again based on the atomic characterization" we suggest " based on the atomic characterization (see (15))"
\item page 10, proof of Lemma 3: Maybe a reference for the density of the Schwartz space in $ S_0 (\mathbb{R}^d)$ could be helpful.
\item page 11, equation 20: the norm of $\sigma_n $ should be $ S_0 '$
\item page 14, equation (29) and 2 and 3 line below it: Since $g$ is chosen to be $g_0$, please replace $g$ by $g_0$ accordingly at 4 occurrences.
\item page 16, line 2: $\gamma_0$ will be introduced in Lemma 4 below.
\item page 17: please explain $w^*$-$w^*$-continuous form
\end{enumerate}
\end{document}

Author Response
Dear reviewers/editors
I have realized practically all the changes suggested.
I have introduced a new Remark 3 (end of section)
Thanks for the suggestion.
HGFei
Reviewer 2 Report
It should be convenient to indicate the end of the proofs.
In p.7, l.2, the symbol γ0 maybe means the Dirac’s delta δ0.
It should be included σ ∈ S′(Rd) in Definition 4.
In the proof of Lemma 3 it should be specified the meaning of f.
Remove the word Check in p. 11, l. 11 and p. 20, l. 3.
In p. 20, the book of Lighthill was first published in 1959.
Maybe, for the sake of completeness, in p. 12 it should be included the definition of a Wiener amalgam space.
Author Response
Thanks for the careful reading. I hope that I have done all the replacements.
The use of the end-of-proof symbol is under discussion with the academic editor.
HGFei *the author
Reviewer 3 Report
The author presents a new description of the “mild distributions” of a Banach Gelfand triplet, which aims to be a universal tool for Fourier Analysis. The description is based on a sequential approach, and it complements other descriptions done by the author and collaborators. I find the paper suitable for publication. I have only two comments:
(1) On page 11, in the third line of the fourth paragraph, where it says “CHECK
Given”, the “CHECK” should be erased.
(2) On the third line of page 20, there seems to be another “check” that needs to be erased.
Author Response
Thanks for pointing out the obvious redundant/left-over "check" symbols.
I have also spotted a few other typos and shortened slightly the abstract so that
everything fits into one page.